# Chemical Composition and Bioactivity of Essential Oil of Ten *Labiatae* Species

**DOI:** 10.3390/molecules25204862

**Published:** 2020-10-21

**Authors:** Mengting Liu, Feiya Luo, Zhixing Qing, Huichao Yang, Xiubin Liu, Zihui Yang, Jianguo Zeng

**Affiliations:** 1Hunan Key Laboratory of Traditional Chinese Veterinary Medicine, Hunan Agricultural University, Changsha 410128, China; lmt19970808@163.com (M.L.); luofeiya1688@163.com (F.L.); qingzhixing@hunau.edu.cn (Z.Q.); isyanghc@163.com (H.Y.); xiubin_liu@hunau.edu.cn (X.L.); 2College of Veterinary Medicine, Hunan Agricultural University, Changsha 410128, China

**Keywords:** *Labiatae* species, essential oil, antioxidant, antibacterial, feed additive

## Abstract

Using antibiotics as feed additives have been successively banned worldwide from 1986; therefore, it is an urgent task to finding safe and effective alternatives. As natural products of plant origin, essential oils (EOs) are an outstanding option due to their reported bioactivity. In this research, ten EOs of *Labiatae* species were extracted by steam distillation and its chemical constituents were identified by gas chromatography-mass spectrometry (GC-MS). A total of 123 chemical compounds, including alkenes, phenols, aldehydes and ketones, were identified. The results of antioxidant activity carried out through DPPH free radical scavenging (DPPH) and ferric reducing antioxidant power (FRAP), showing that EOs of *Ocimum basilicum* Linn. (ObEO), *Thymus mongolicus* Ronn. (TmEO), *Origanum vulgare* Linn. (OvEO) and *Mosla chinensis* Maxim. (McEO) have strong antioxidant activities. Their 50%-inhibitory concentration (IC_50_) value was <1.00, 1.42, 1.47 and 1.92 μg/mL, respectively; and their FRAP value was 1536.67 ± 24.22, 271.84 ± 4.93, 633.71 ± 13.14 and 480.66 ± 29.90, respectively. The results of filter paper diffusion showing that McEO, OvEO and TmEO inhibition zone diameter (IZD) are all over 30 mm. The results of two-fold dilution method showed that McEO, OvEO and TmEO have strong antibacterial activities against *Staphylococcus aureus (S. aureus)* and their minimal inhibitory concentrations (MIC) value was 1 μL/mL, 2 μL/mL, and 2 μL/mL, respectively. In conclusion, the results in this work demonstrate the possibility for development and application of EOs as potential feed additives.

## 1. Introduction

Antibiotics, the development of which dates back to 1929, have been widely used as feed growth-promoting additives (AGPs) for a long time. Nevertheless, antibiotics were constantly and globally banned as feed additives from 1986, due to its drug resistance and safety hazards [1]. AGPs were prohibited using as commercial feed additives from July 1, 2020, in China; therefore, the product which can replace the AGPs become a pressing need in the breeding industry. Meanwhile, the search for relevant alternatives from plants is a focal research point due to its low toxicity, no drug resistance, no residues and no pollution [2]. Plant extracts with antioxidant activity, antibacterial activity and regulating intestinal health, make it become potential alternatives of AGPs, are getting more and more attention [3,4].

Essential oils (EOs), important secondary metabolites of plants contain several chemical classes of compounds, which are volatile at normal temperature, immiscible with water and obtained by steam distillation from plants [5,6]. EOs presented more biological properties, such as antibacterial activity [7,8,9], antioxidant activity [10,11] and regulating intestinal function [12], and have been widely used in pharmaceutical [9], agriculture [13,14], food [15] and cosmetic industries [16]. EOs have also attracted increasing attention from the market and researchers in the breeding industry. The use of EOs as alternatives to antibiotics has always had great expectation, and the search for low-cost and effective essential oils from forage plants as feed additives is a hot topic of research.

*Labiatae* species are an important source of EOs, with 10 subfamilies, more than 220 genera, and about 3500 species distributed throughout the world. There are 99 genera and 808 species in China, and many of them have a wide range of pharmacological activities. In particular, studies have shown that EOs of the *Labiatae* family have significant bioactivity and have the potential to be developed as an alternative to antibiotics [17].

In this research, ten EOs of *Labiatae* species including *Thymus mongolicus* Ronn., *Mosla chinensis* Maxim., *Origanum vulgare* Linn., *Rosmarinus officinalis* Linn., *Ocimum basilicum* Linn., *Mentha haplocalyx* Briq., *Pogostemon* cablin (Blanco) Benth., *Mentha spicata* Linn., *Salvia officinalis* Linn., *Perilla frutescens* (Linn.) Britt. were chosen as the research objects to analysis its chemical composition and biological activity. EOs of ten *Labiatae* species were extracted, with the chemical constituents were analyzed and identified through GC-MS, and their antioxidant and antibacterial activities were evaluated using DPPH and ferric reducing antioxidant power (FRAP).

## 2. Results and Discussion

### 2.1. Extraction and Yield

10 EOs of *Labiatae* species were obtained by hydrodistillation, with a yield of 0.75, 1.73, 0.30, 1.90, 0.13, 0.82, 0.52, 0.42, 0.25 and 1.99% respectively (Figure 1). The highest (1.99%) belonged to *Perilla frutescens* (Linn.) Britt. (PfEO), while the lowest (0.13%) was recorded for the *Ocimum basilicum* Linn. (ObEO).

### 2.2. Chemical Composition

The chemical constituents of 10 EOs were identified by GC-MS combined with the database of NIST.17 (Table 1). Regarding the chemical profile of the EOs, a total of 123 components were identified. The results show that significant differences in the chemical composition of 10 EOs. The relative content of each compound was determined using peak area normalization. The major components of each EOs displayed in Table 1, and the total compound content was also identified. The largest amounts of 10 EOs are D-carvone (71.1%) and menthol (69.05%), followed by patchouli alcohol (50.52%), perillaketone (35.56%), tanacetone (27.99%), thymol (23.7%), linalool (26.65%), α-pinene (26.46%).

38 compounds were identified in TmEO, accounting for 90.51% of the total EO, The main chemical constituents of TmEO are alkenes and phenols, such as γ-terpinene (16.42%), *p*-cymene (21.17%), thymol (23.70%). 22 compounds were identified in McEO, accounting for 94.25% of the total EO, with γ-terpinene (15.14%), *p*-cymene (19.52%), thymol (26.59%), alkenes and phenols compounds were dominated. 36 compounds were identified in OvEO, accounting for 89.87% of the total EO, the main chemical constituents are thymol (14.64%) and carvacrol (20.82%), which belonged to phenols. Thymol was the common component of TmEO, McEO and OvEO, this is consistent with the literature reported previous [18,19,20]. 34 compounds were identified in RoEO, accounting for 86.68% of the total EO, the main chemical components are eucalyptol (11.31%), verbenone (16.56%) and α-pinene (26.46%), monoterpenoid ketones and alkenes were dominated. While the research of Bouyahya A [21] showed the main compounds were α-pinene, 1, 8-eucalyptol and menthol, which is a little bit different from our results. It is probably due to different origins and extraction methods. 43 compounds were identified in ObEO, accounting for 92.06% of the total EO, linalool (26.65%), *trans*-α-bergamotene (14.16%), eugenol (10.27%), which belonged to alcohols, alkenes and phenols are main chemical consitituents. Because of different origins, the reported by Kathirvel [22] displayed that methyl cinnamate and Linalool were the main components. 23 compounds were identified in MhEO, accounting for 97.02% of the total EO, menthol (69.05%) was the most abundant in MhEO. 19 compounds were identified in PcEO, accounting for 81.58% of the total EO, the main component of PcEO was the patchouli alcohol (50.52%). 30 compounds were identified in MsEO, accounting for 96.85% of the total EO, the main chemical components are D-carvone (71.10%), which is consistent with previous reports [23,24,25]. 35 compounds were identified in SoEO, accounting for 97.24% of the total EO, tanacetone (27.99%), camphor (16.21%) are main chemical consitituents, ketones and terpene were dominated. 31 compounds were identified in PfEO, accounting for 94.22% of the total EO, the main chemical components are perillaketone (35.56%), isoegomaketone (20.40%), which belonged to ketones. However, the results were different from previous reports [26,27], which may be caused by different habitats of plants.

Statistical analysis of the chemical composition of 10 EOs shows that alkenes, phenols, aldehydes and ketones were main chemical composition, with a small number of alcohols and their oxides. Monoterpene hydrocarbons and phenolic acid compounds, which are closely related to the biological activities of EO, are dominant.

### 2.3. Antioxidant Activity

Antioxidant activity is a complex process usually occurring through several mechanisms, the evaluation of the antioxidant activity should be carried out by more than one test method [28]. In this study, two antioxidant assays, 1,1-diphenyl-2-picrylhydrazyl (DPPH) free radical scavenging activity and ferric reducing antioxidant power (FRAP) were applied to accurately evaluate the antioxidant properties of 10 EOs.

#### 2.3.1. DPPH Free Radical Scavenging Activity

The antioxidant activity results of 10 EOs were determined by DPPH (Figure 2), which shows that there is an obvious dose-effect relationship between DPPH radical clearance rate and concentration of 10 EOs. The IC_50_ values were range from 0 to 49.74 μg/mL, with ObEO (<1 μg/mL), TmEO (1.42 μg/mL), OvEO (1.47 μg/mL), McEO (1.92 μg/mL), PfEO (3.77 μg/mL), SoEO (11.86 μg/mL), MsEO (13.34 μg/mL), RoEO (20.36 μg/mL), MhEO (23.95 μg/mL) and PcEO (49.74 μg/mL) (Table 2).

#### 2.3.2. Ferric Reducing Antioxidant Power (FRAP)

The determination of total antioxidant activity with the FRAP method is the antioxidant can turn Fe^3+^-TPTZ complex to blue-purple Fe^2+^-TPTZ under acidic conditions, with the absorbance of Fe^2+^-TPTZ determined by Elisa. The change of absorbance is proportional to the content of the reduced substance, and the maximum absorption is reached at 593 nm, so it can be used as an indicator when evaluating the total antioxidant activity of sample [29]. The FRAP values of 10 EOs were calculated according to the standard curve (Appendix A), and the results are showed in Table 2 followed the crescent order as ObEO > OvEO > McEO > TmEO > PfEO > PcEO >MsEO > MhEO > RoEO > SoEO.

The results of DPPH and FRAP showed that 10 EOs of *Labiatae* species had strong antioxidant activity. Among the *Labiatae* species, ObEO with IC_50_ < 1 μg/mL and the highest FRAP value illustrate the strongest antioxidant activity, and the McEO, TmEO and OvEO own better antioxidant activity than other EOs. ObEO contains a large amount of linalool and eugenol, which may correspond to the antioxidant effect, is consistent with the previous reports [30]. Because of the thymol, which is the common component of TmEO, McEO and OvEO among the remaining nine EOs, has a certain amount of antibacterial and antioxidant activities, and even have been widely used in the breeding industry. Therefore, TmEO, McEO and OvEO show better antibacterial and antioxidant activities.

In a word, the results indicate that ten EOs of *Labiatae* species, especially for ObEO, could be potential sources of natural antioxidant feed additives, and expected to be used as feed additives in breeding industry.

### 2.4. Antibacterial Activity

The antibacterial activity of 10 EOs against four microorganisms was examined qualitatively (inhibition zone diameter, IZD), quantitatively (minimal inhibitory concentrations (MIC) and with minimal bactericidal concentration (MBC)). IZD tests carried out by the filter paper diffusion method, MIC and MBC were calculated using the two-fold dilution method.

#### Results of IZD, MIC and MBC

McEO, OvEO, TmEO had the most apparent antibacterial effect on *Staphylococcus aureus (S. aureus)*, with the IZD values are more than 30 mm (Table 3). They also showed strong inhibitory effects on *Escherichia coli (E. coli)*, *Bacillus subtilis (B. subtilis),* and *Salmonella enteritidis (S. enteritidis)*. TmEO presented better antibacterial effect on *S. aureus*, compared with the results from Zhang et al. [31], in which the IZD value was 19.1 mm, although worse on *B. subtilis and E. coli,* with the IZD value of 34.5 and 15.0 mm respectively. The results of McEO is better than Li et al. [19], which the IZD value was 14.7 mm (*S. aureus*) and 8.0 mm (*E. coli*). Compared to the work of Assiri et al. [32], the IZD value of OvEO against *S. aureus* is consistent, while the inhibitory effect of OvEO against *E. coli*. and *S. enteritidis* is unsatisfactory.

The MIC values of 10 EOs are different on the tested microorganisms (Table 3).This indicates that the EOs’ antibacterial activity on four microorganisms is not regular, such as PfEO and MhEO to *E. coli*, MsEO to *B. subtilis* and *S. enteriditis*, SoEO against *S. aureus* and *E. coli*. PcEO even had no MIC on all four tested microorganisms. Nevertheless, McEO, OvEO and TmEO had obvious antibacterial effect on *S. aureus,* with the MIC value were 1, 2, 2 μL/mL respectively, is consistent with the IZD results. The MIC value for TmEO to *B. subtilis*, *S. aureus* and *E. coli* is higher than the results of Zhang et al. [31], while nearly the same with the results of Niu et al. [33]. The differences of thyme phenol content which was the main ingredient in TmEO maybe the main reason. McEO’s MIC value is much better than Li et al. [19], of which the MIC value is 62.5 μg/mL (*S. aureus*) and 250 μg/mL (*E. coli*). Dutra et al. [34], in their research, evaluated the antibacterial activity of OvEO against *S. aureus* and *E. coli* with the both MIC value of 12.5 μg/mL.

The inhibitory effect of other seven EOs on the four microorganisms are weak, SoEO and PfEO had no obvious inhibitory effect on the four tested microorganisms. The RoEO’s antibacterial activity, evaluated by Hussain et al. [35], is more desirable relative to our work. Similarly, it is the same situation to the ObEO compared to the results of Saha et al. [36] Though Delamare et al. [37] have evaluated the antibacterial activity of SoEO, there is no practical significance of the MIC values against *E. coli, B. subtilis* and *S. aureus,* due to the results were 5–10 mg/mL. In addition, there were very few reports on antibacterial activity of MhEO, PcEO and PfEO.

According to the results of MBC (Table 3), it is obvious that McEO, OvEO, TmEO own strong inhibitory effects to the four microorganisms, with disappointing results of the rest seven EOs. It is interesting that most of the reports on antibacterial activity of EOs focus on the determination of IZD and MIC, without MBC. Research on antibacterial activity focus more on inhibition than on sterilization may probably the reason. Meanwhile, plant essential oils are a potentially useful source of antibacterial compounds. The results of different studies are difficult to compare, most probably because of the different test methods, bacterial strains and sources of antibacterial samples used.

EOs have been used as an alternative to antibiotic growth promoter on intestinal health, immune response and antioxidant status in broiler chickens Chowdhury et al. [38], due to its antibacterial activity. Essential oil and aromatic plants as feed additives in non-ruminant nutrition is also well-known for a long time Zeng et al. [39]. However, the quality of EOs from different origin is uneven during the practical applications, and it is difficult to identify the EOs products which replaced by synthetic product. Therefore, the quality of EOs is out of control, and we suggest utilizing the specific chromatogram technology to identify the different EOs.

### 2.5. Antibacterial Stability of Essential Oils

The antibacterial experiments were tested under different temperature, pH and UV irradiation time, and the antibacterial results were compared with those under normal conditions.

#### 2.5.1. Effects of Different pH on Antibacterial Stability of EOs

EOs had different effects on the antibacterial activities of four tested microorganisms under different pH (Figure 3 and Appendix A). The results showed that TmEO, McEO and OvEO of these ten EOs had stronger inhibitory effect on the four tested microorganisms than the other seven, and there were no significant differences on the inhibitory effect of the other seven EOs on *E. coli*, *B. subtilis*, and *S. enteriditis* within different pH. The effect of pH on the inhibitory effect of essential oils against *S. aureus* is diverse, pH = 8 was the best condition to MhEO, MsEO and PfEO, while ObEO and PcEO performed best at pH = 6, and three conditions is equally effective for RoEO and SoEO. According to the reported literature, the evaluation of EOs’ pH on antibacterial activity is rarely reported. Most of the reports usually combined EOs with other substance as products, such as gum arabic (Niu et al. [33]), alginate (Sarengaowa et al. [40]) and so on.

#### 2.5.2. Effects of Different Temperature on Antibacterial Stability of EOs

The inhibitory effects on *S. aureus* of 10 EOs can be divided into two categories, one is stable team consist of TmEO, McEO and OvEO, which the inhibitory effects was stable with the temperature changing, and the rest EOs are the other team, which the inhibitory effects improved as the increasing of temperature (Figure 4 and Appendix A). Conversely, the inhibitory effects on *E. coli*, *B. subtilis* and *S. enteriditis* is opposite to *S. aureus*. The temperature only obviously affects TmEO, McEO and OvEO, with no significant fluctuations on the rest seven EOs. Though the results shown that the EOs own thermal stability, lots of reported work have directly choose 37 °C as experimental temperature [19,31,32,33].

#### 2.5.3. Effects of UV Irradiation Time on Antibacterial Stability of EOs

The antibacterial activity of SoEO and ObEO to the four tested microorganisms had no significant effect under UV irradiation. Nevertheless, UV irradiation enhanced the antibacterial activity of PfEO, MhEO, MsEO and PcEO to *S. aureus*, with no apparent impact on the other three tested microorganisms. The enhancement of antibacterial activity of the other 6 EOs under UV irradiation is evident (Figure 5 and Appendix A). The composition of EOs may change under UV irradiation, which may be one of the factors that lead to the enhancement of the antibacterial activity of EOs.

In a word, part of Eos’ antibacterial activity was greatly diminished under UV irradiation, while rest of them were mainly enhanced. Therefore, plant EOs can also achieve a good antibacterial effect under this condition.

The results show that the antibacterial activity of 10 EOs has small amplitude fluctuation under different temperature range (50~90 °C), different pH (6~8) and different UV irradiation time (10~30 min), but there is no significant difference compared with the normal control group. The antibacterial activity of 10 EOs generally remained stable under different temperature, pH and UV irradiation time.

## 3. Materials and Methods

### 3.1. Plant Materials Collection and EOs Extraction

*Thymus mongolicus* Ronn. (Shandong, Qingdao), *Mosla chinensis* Maxim. (Hunan, Xinning), *Origanum vulgare* Linn. (Shandong, Qingdao), *Rosmarinus officinalis* Linn. (Hunan, Changsha), *Ocimum basilicum* Linn. (Shandong, Qingdao), *Mentha haplocalyx* Briq. (Shandong, Qingdao), *Pogostemon cablin* (Blanco) Benth. (Shandong, Qingdao), *Mentha spicata* Linn. (Shandong, Qingdao), *Salvia officinalis* Linn. (Shandong, Qingdao), *Perilla frutescens* (Linn.) Britt. (Shandong, Qingdao), were collected from different regions of China. They were unambiguously authenticated by Prof Guangmin Yang (Hunan University of Chinese Medicine, Hunan, China).

The EOs of 10 plants were extracted using the steam distillation [41]. The samples were dried in a ventilated place, and cut into 1 cm pieces and separately hydro-distilled for 4 h (200 g samples in 1200 to 1600 mL of distilled water) in a round-bottom flask equipped an EO tester and condensation reflux tube, of which filled with distilled water. The EOs dried with anhydrous sodium sulfate and collected in a brown bottle, stored at 4 °C before analysis. The yields of the EOs were calculated by the Equation (1).
(1)$$\mathrm{Yield}\;\mathrm{of}\;\mathrm{EO}\;=\;\frac{\mathrm{Volume}\;\mathrm{of}\;\mathrm{EO}(g)}{\mathrm{Volume}\;\mathrm{of}\;\mathrm{sample}(g)}\;\times \;100\%$$

### 3.2. Identification of the Chemical Components of the EOs

The GC/MS analysis was carried out using splitless injection mode on a QP2010 gas chromatograph-mass spectrometer instrument (Shimadzu, Yubinbang, Japan) equipped with a CD-WAX chromatographic column (30 m, 0.25 mm, 0.25 μm film thicknesses, ANPEL, Shanghai, China), and data analysis through GC/MS solution chromatography workstation and NIST.17 mass spectrometry database (Shimadzu, Yubinbang, Japan). Helium was used as carrier gas at a flow rate of 1.0 mL/min, splitless. The oven temperature was programmed from low to high designed at a specified speed (Appendix A) and injector heater 240 °C. The mass-spectrometer was accomplished in the range of 28–500 m/z in the EI-mode at 70 eV and ion source temperature was set to 200 °C. The components of EOs were identified by matching their recorded mass spectra with the data bank mass spectra (NIST 17). For each compound on the gas chromatogram, the percentage of peak area relative to the total peak area of all compounds was determined and reported.

### 3.3. Determination of Antioxidant Activity

#### 3.3.1. Antioxidant Activity Determined by DPPH

The ability to scavenge DPPH free radicals was evaluated according to the procedure reported with some modifications [42,43]. At first, 2.0 mL Eos (Homemade, Changsha, China) were blended at chosen concentrations (1, 5, 10, 20, 40 and 80 μg/mL) with 0.2 mM ethanol DPPH solution (2 mL, Beyotime, Shanghai, China). While the reaction mixture without any sample was used as a negative control (A_0_). Reaction mixtures were incubated in the dark at room temperature for 30 min, the absorbance at 517 nm was recorded using a microplate reader (NanoQuant infinite M200Pro TECAN, Tecan, Hombrechtikon, Switzerland). The inhibition percentage was plotted versus the sample concentration and 50% of the inhibitory concentration (IC_50_) of the DPPH values was defined by linear regression analysis. Ascorbic acid (VC) of different concentrations (1–80 μg/mL) and *N*-phenylacetyl-l-glutamine (PG) were used as a positive control, The scavenging activity percentage against DPPH radicals was calculated as follows the Equation (2).
SR% = (1 − (A_i_ − A_j_)/A_0_) × 100 %(2)
where, A_i_ is the absorbance of the sample at 30 min; A_0_ is the absorbance of reaction mixture without any sample at 30 min; A_j_ is the absorbance of the ethanol (100%) and EOs.

#### 3.3.2. Antioxidant Activity Determined by FRAP

The FRAP assay was evaluated according to Chen with some modifications [44]. FRAP reagent (a kind of rapid test kits, Shanghai Beyotime Biotechnolgy Co., LTD, Shanghai, China). According to the operating instructions, 180 μL FRAP reagent was mixed at 37 °C with each test sample solution (5 μL) of different concentration (TmEO (2 μg/mL), McEO (2 μg/mL), OvEO (2 μg/mL), RoEO (20 μg/mL), ObEO (1 μg/mL), MhEO (20 μg/mL), PcEO (20 μg/mL), MsEO (20 μg/mL), SoEO (20 μg/mL), PfEO (20 μg/mL)) in 96-well plates. After 3–5 min, the colored products were then taken at 593 nm. Results were expressed as trolox equivalent antioxidant capacity. The standard curve was determined by the FeSO_4_ standard solution. Trolox as the positive control the total antioxidant capacity of the sample was expressed by the FRAP value. The value of 1 FRAP is equivalent to 1 mM FeSO_4_.

### 3.4. Determination of Antibacterial Activity

#### 3.4.1. Test Microorganisms

The microbial strains applied in this investigation were Escherichia coli (*E. coli*), Staphylococcus aureus (*S. aureus*), Salmonella enteritidis (*S. enteriditis*), Bacillus subtilis (*B. subtilis*), which were provided by China Center for Type Culture Collection (CCTCC, Wuhan, China). The strains were inoculated in liquid lysogeny broth (LB) medium and cultured (37 °C) for 24 h after activated, then diluted with new liquid LB mediums to a suspension liquid (10^7^ CFU/mL).

#### 3.4.2. Inhibition Zone Diameter

The filter paper diffusion method was used to obtain the inhibition zone [45]. EOs (2 μL) were absorbed on the filter paper (*d* = 6 mm), meanwhile, applied to the Muller Hinton Agar plates (pH = 7) with the proper bacteria solution (100 μL, 10^6^ CFU/mL, manufacturer, city, country). Paper contained EOs in duplicate and 1 piece of paper contained sterile water were put on a plate, Chlortetracycline Hydrochloride was used as a positive control, then the plates were inoculated at 37 °C for each strain for 12 h. Crossing method was used to determine the inhibition zone diameter [46].

#### 3.4.3. Minimal Inhibitory Concentrations (MIC) and Minimal Bactericidal Concentration (MBC)

The MIC and MBC values of ten EOs and CTC against two Gram-positive bacteria and two Gram-negative bacteria were evaluated using two-fold dilution method as reported earlier [47,48]. Ten EOs were diluted into 320 μL/mL with 1% tween 80 aqueous solution. Take 11 sterile tubes numbered 1–11, 1.6 mL LB medium was added to the tube NO.1, while NO. 2–11 were added with 1.0 mL, respectively. Then 0.4 mL EOs was added into tube NO.1, following aspirate 1.0 mL of the mixed solution to the tube NO.2. Similarly, aspirate 1.0 mL of the mixed solution in tube NO.2 to the tube NO.3, and the same operation until NO.9. The tube NO.10 and NO.11 were used as sample blank and matrix blank, with no addition of mixed solution from previous tube. To keep the volume of solution consistent in all tubes, discard 1.0 mL of the mixed solution from tube NO.9.

Then, 1.0 mL bacterial suspension (10^7^ CFU/mL) was added to the tube NO.1-9, make the concentration of EOs was 0.125 to 32 μL/mL and a concentration of bacterial was 10^6^ CFU/mL. The tubes were inoculated at 37 °C for each strain for 24 h, and then each tube (100 μL) was pipetted to the agar plate, inoculated at 37 °C for 12 h. The MIC of EOs were specified as the lowest concentration showing no perceptible microbial growth. Chlortetracycline Hydrochloride was used as a positive control. MBC value was determined by plates that showed no growth and incubating at 37 °C for another 12 h. The lowest concentration that disclosed no visible growth of bacteria was deemed as MBC.

#### 3.4.4. Antibacterial Stability of EOs

The experiment used 0.1 mol/L NaOH and 0.1 mol/L HCl to adjust the pH of the medium to 6.0, 7.0 and 8.0, respectively, then test the antibacterial activities of EOs under different pH. Ten EOs were placed in 1.5 mL tube and irradiated by UV light for 0, 10, 20 and 30 min on the benchtop respectively, then test the antibacterial activities of EOs. Ten EOs were put into 10 amber bottles respectively, and then placed in water bath pot with different temperatures (37, 50, 70 and 90 °C), then test the antibacterial activities of EOs after 30 min.

### 3.5. Statistical Analysis

All analyses were performed in triplicate and the results expressed as the mean standard deviation (SD). For the data with normal distribution, an analysis of variance (ANOVA) was performed, considering the compound and its concentration as fixed factors, while the inhibition percentage as the dependent variable. The significance criterion was set to *p* < 0.05.

## 4. Conclusions

Chemical composition and bioactivity of essential oil of ten *Labiatae* species have been investigated in this research. The yield of these EOs is satisfactory, which make it possible to develop into products in economic value. A range of chemical compounds have been identified from these ten EOs, with alkenes, phenols, aldehydes and ketones were major categories. The results of DPPH and FRAP have proved that ObEO own the best antioxidant activity, and followed OvEO, McEO and TmEO. Meanwhile, OvEO, McEO and TmEO also presented desirable antibacterial activity. ObEO, McEO, TmEO, OvEO get the most potential of developing into a new kind of feed additives, and also should have broad development potential in food, feed, cosmetics and other industries.

## Figures and Tables

**Figure 1 molecules-25-04862-f001:**
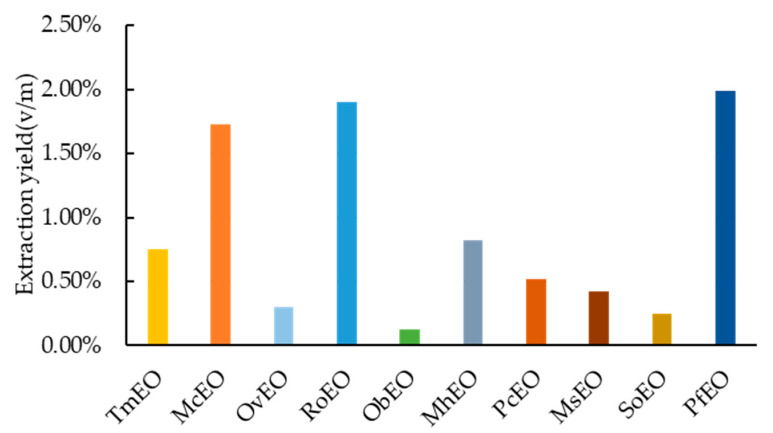
Extraction yield of ten essential oils (EOs). TmEO: *Thymus mongolicus* Ronn.; McEO: *Mosla chinensis* Maxim.; OvEO: *Origanum vulgare* Linn; RoEO: *Rosmarinus officinalis* Linn.; ObEO: *Ocimum basilicum* Linn.; MhEO: *Mentha haplocalyx* Briq.; PcEO: *Pogostemon cablin* (Blanco) Benth.; MsEO: *Mentha spicata* Linn.; SoEO: *Salvia officinalis* Linn.; PfEO: *Perilla frutescens* (Linn.) Britt.

**Figure 2 molecules-25-04862-f002:**
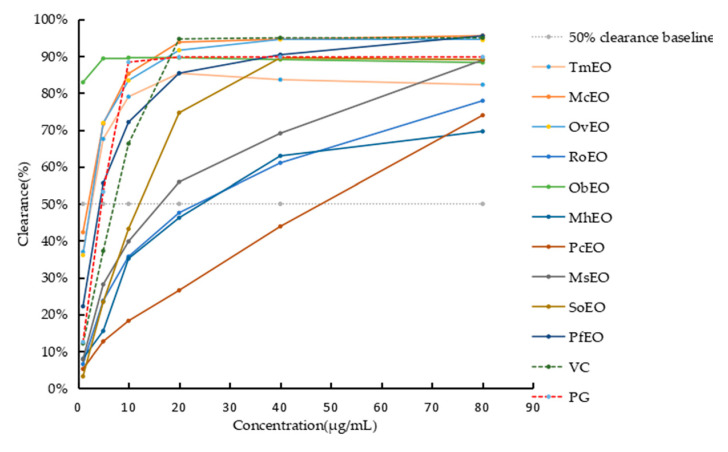
Clearing-concentration relationship of 10 EOs, ascorbic acid (VC) and *N*-phenylacetyl-l-glutamine (PG).

**Figure 3 molecules-25-04862-f003:**
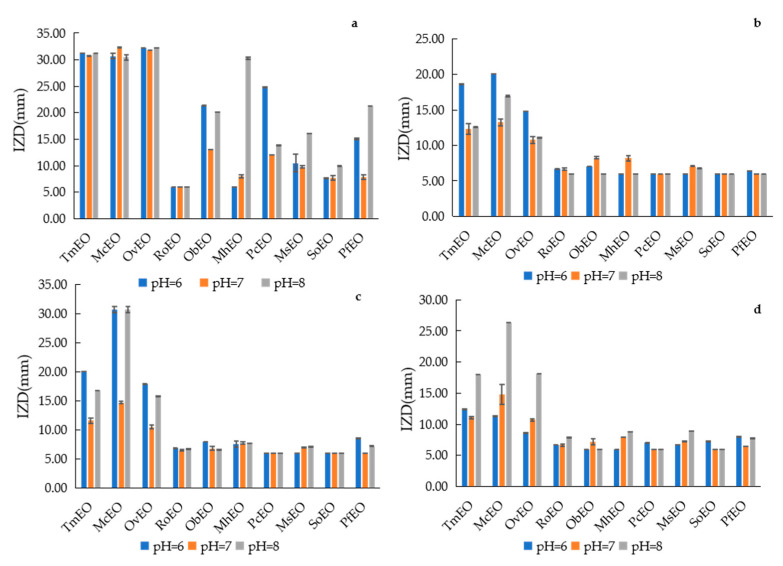
IZD of EOs under different pH. (**a**) *S. aureus*, (**b**) *E. coli*, (**c**) *B. subtilis*, (**d**) *S. enteriditis*.

**Figure 4 molecules-25-04862-f004:**
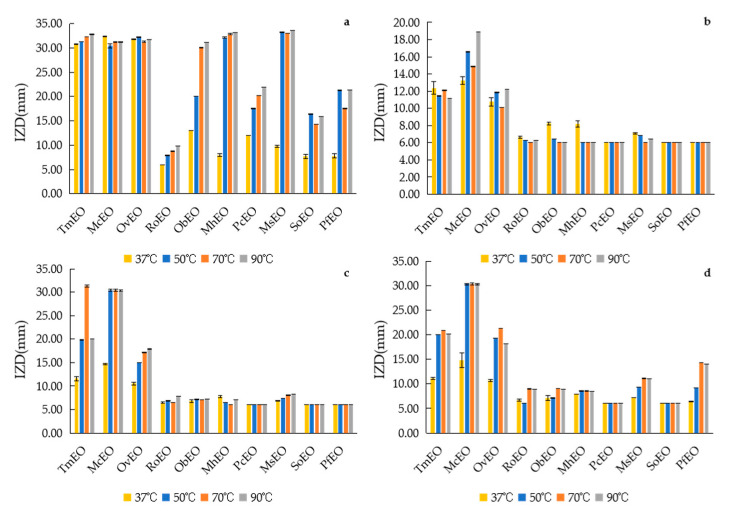
IZD of EOs at different temperature. (**a**) *S. aureus*, (**b**) *E. coli*, (**c**) *B. subtilis*, (**d**) *S. enteriditis*.

**Figure 5 molecules-25-04862-f005:**
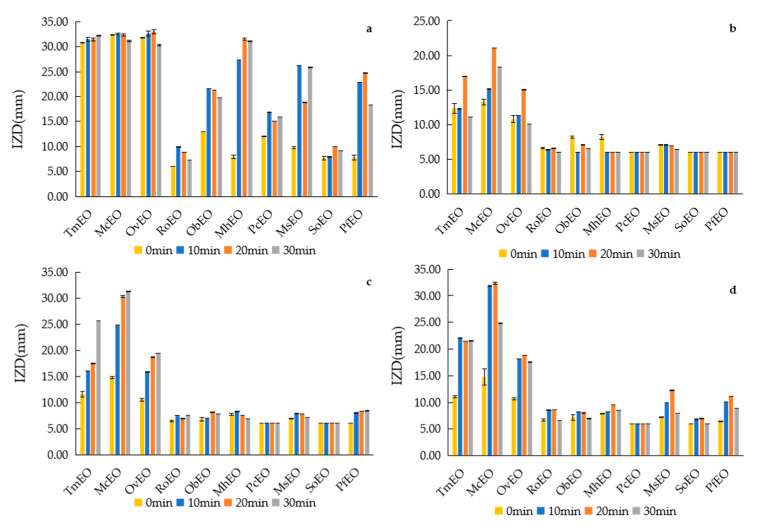
IZD of EOs under different UV irradiation time. (**a**) *S. aureus*, (**b**) *E. coli*, (**c**) *B. subtilis*, (**d**) *S. enteriditis*.

**Table 1 molecules-25-04862-t001:** Percentage composition of the EOs from 10 plants.

No	Molecular Formula	Compounds	Relative Peak Area (%)									
			TmEO	McEO	OvEO	RoEO	ObEO	MhEO	PcEO	MsEO	SoEO	PfEO
1	C_10_H_16_	α-Pinene	0.46	0.13	-	26.46	0.09	0.4	-	0.58	1.93	-
2	C_10_H_16_	(−)-Camphene	-	-	-	3.53	-	-	-	-	-	-
3	C_10_H_16_	α-thujene	1.14	0.26	-	-	-	-	-	-	-	-
4	C_10_H_16_	Camphene	0.49	0.1	-	-	-	-	-	0.03	3.63	-
5	C_10_H_16_	β-Pinene	0.2	-	0.4	1.09	0.22	0.31	-	0.59	3.65	-
6	C_10_H_14_	2,4(10)-Thujadiene	-	-	-	0.72	-	-	-	-	-	-
7	C_10_H_16_	Sabinene	0.08	-	-	-	0.08	0.08	-	0.25	0.19	-
8	C_10_H_16_	β-Myrcene	0.87	2.18	-	0.83	-	0.27	-	1.59	0.6	-
9	C_10_H_16_	α-terpinene	2.44	3.19	0.84	0.44	-	-	-	-	0.11	-
10	C_15_H_24_	β-Patchoulene	-	-	-	-	-	-	1.23	-	-	-
11	C_15_H_24_	α-Guaiene	-	-	-	-	-	-	6.09	-	-	-
12	C_15_H_24_	γ-Gurjunene	-	-	-	-	-	-	0.49	-	-	-
13	C_10_H_16_	d-Limonene	0.3	0.26	0.06	0.85	0.07	1.53	-	9.64	0.49	1.38
14	C_10_H_18_O	Eucalyptol	-	-	0.31	11.31	4.48	-	-	0.64	5.11	-
15	C_10_H_16_	trans-β-Ocimene	-	-	-	-	-	-	-	0.48	0.03	-
16	C_10_H_16_	allo-Ocimene	-	-	-	-	-	-	-	0.16	-	-
17	C_10_H_16_	β-phellandrene	-	0.25	-	-	-	-	-	-	-	-
18	C_10_H_16_	γ-Terpinene	16.42	15.14	6.14	1.11	0.09	-	-	-	0.46	-
19	C_10_H_16_	β-Ocimene	-	-	-	-	0.71	-	-	-	-	-
20	C_10_H_14_	*p*-Cymene	21.17	19.52	14.11	1.76	-	-	-	-	0.25	-
21	C_10_H_16_	(+)-4-Carene	0.09	-	-	-	-	-	-	-	0.23	-
22	C_10_H_16_O	Tanacetone	-	-	-	-	-	-	-	-	27.99	-
23	C_10_H_16_O	Thujone	-	-	-	-	-	-	-	-	5.24	-
24	C_10_H_16_	Terpinolene	-	0.18	-	0.85	0.13	-	-	-	-	-
25	C_6_H_12_O	3-Hexen-1-ol	-	-	-	-	0.33	-	-	-	-	-
26	C_8_H_18_O	3-Octanol	0.27	-	2.05	-	-	0.6	-	0.31	-	0.1
27	C_10_H_14_O	Perillene	-	-	-	-	-	-	-	-	-	1.62
28	C_10_H_18_O	l-Menthone	-	-	-	-	-	12.21	-	-	-	-
29	C_10_H_18_O	(+)-Isomenthone	-	-	-	-	-	3.07	-	-	-	-
30	C_8_H_16_O	1-Octen-3-ol	2.97	-	1.66	0.02	0.05			0.14	0.08	0.18
31	C_10_H_18_O	*Cis*-β-Terpineol	-	-	-	-	0.19	-	-	-	-	-
32	C_10_H_16_O	(+)-2-Bornanone	-	-	-	4.08	-	-	-	-	-	-
33	C_10_H_16_O	Isopinocamphon	-	-	-	1	-	-	-		0.06	-
34	C_10_H_14_O	Pinocarvone	-	-	-	0.41	-	-	-	-	-	-
35	C_10_H_16_O	Camphor	0.1	-	0.57	-	0.67	-	-	-	16.21	
36	C_15_H_24_	(−)-β-Bourbonene	-	-	0.23	-	0.14	0.16	-	1.21	-	0.07
37	C_12_H_22_O_2_	l-Menthyl acetate	-	-	-	-	-	3.73	-	-	-	-
38	C_10_H_18_O	Linalool	2.97	0.06	0.14	2.99	26.65	-	-	0.49	0.37	2.97
39	C_10_H_16_O	2-methyl-5-(1-methylethenyl)-Cyclohexanone	-	-	-	-	-	-	-	2.76	-	-
40	C_11_H_16_O	2-Isopropyl-1-methoxy-4-methylbenzene	-	-	11.34	-	-	-	-	-	-	-
41	C_7_H_8_O_2_	5-methyl-2-acetyl-Furan	-	-	0.23	-	-	-	-	-	-	-
42	C_10_H_20_O	Menthol	-	-	0.97	-	-	69.05	-	-	-	-
43	C_10_H_12_O	Estragole	-	-	0.31	-	1.32	-	-	-	-	-
44	C_15_H_24_	*cis*-Muurola-4(15),5-diene	-	-	-	-	0.29	-	-	-	-	-
45	C_15_H_24_	γ-Muurolene	-	-	0.34	-	4.14	-	-	-	-	-
46	C_12_H_20_O_2_	Bornyl acetate	0.26	-	-	1.85	2.28	-	-	-	0.73	-
47	C_11_H_16_O	2-Isopropyl-5-methylanisole	3.17	-	5.6	-	-	-	-	-	-	-
48	C_15_H_24_	Caryophyllene	0.03	-	-	0.64	-	0.44	-	-	1.81	10.21
49	C_10_H_14_O_2_	Elsholtzia ketone	-	-	-	-	-	-	-	-	-	0.59
50	C_10_H_20_O	Neoisomenthol	-	-	-	-	-	1.63	-	-	-	-
51	C_10_H_16_O	Pulegone	-	-	-	-	-	0.75	-	-	-	-
52	C_10_H_18_O	Lavandulol	-	-	-	-	-	0.37	-	-	-	-
53	C_10_H_18_O	α-Terpineol	-	-	-	-	-	0.11	-	-	-	-
54	C_10_H_18_O	Terpinen-4-ol	-	-	-	1.21	0.76	-	-	0.06	0.63	-
55	C_10_H_16_O	Hotrienol	-	-	-	0.02	0.07	-	-	-	-	-
56	C_10_H_18_O	*p*-menth-1(7)-en-8-ol	-	-	-	0.32	0.27	-	-	-	-	-
57	C_10_H_14_O	Verbenone	-	-	-	16.56	-	-	-	-	-	-
58	C_12_H_20_O_2_	Nerol acetate	-	-	-	0.57	-	-	-	-	-	-
59	C_10_H_20_O	(*S*)-3,7-dimethyl-7-Octen-1-ol	-	-	-	0.61	-	-	-	-	-	-
60	C_15_H_24_	Humulene	0.25	3.74	-	-	-	-	0.16	0.05	6.44	1.14
61	C_15_H_24_	(*E*)-β-Famesene	-	-	-	-	-	-	-	0.55	-	0.23
62	C_15_H_24_	*cis*-β-farnesene	0.05	-	-	-	2.27	-	-	-	-	-
63	C_10_H_18_O	Borneol	2.33	0.1	1.34	-	0.03	-	-	-	1.89	-
64	C_15_H_24_	(+)-δ-Cadinene	-	-	-	-	-	-	-	-	0.04	0.02
65	C_10_H_16_O	Piperitone	-	-	0.61	-	-	1.6	-	-	-	2.94
66	C_10_H_14_O	d-Carvone	-	-	0.89	-	-	-	-	71.1	-	-
67	C_10_H_18_O	Dihydrocarveol	-	-	0.34	-	-	-	-	1.14	-	-
68	C_12_H_18_O_2_	*cis*-Carvyl acetate	-	-	-	-	-	-	-	0.87	-	-
69	C_15_H_22_	*cis*-Calamenene	-	-	-	-	-	-	-	0.34	-	-
70	C_15_H_24_	Germacrene D	0.77	-	-	-	4.35	-	-	-	-	0.76
71	C_15_H_24_	α-Bulnesene	-	-	-	-	1.74	-	5.68	-	-	-
72	C_15_H_26_O	Ledol	-	-	-	-	-	-	1.22	-	-	-
73	C_10_H_14_O	(−)-Carvone	0.03	-	-	-	-	-	-	-	-	0.22
74	C_15_H_24_	(*Z*,*E*)-α-Farnesene	-	-	-	-	-	-	-	-	-	4.44
75	C_15_H_24_	γ-Elemene	0.03	-	-	-	0.08	-	-	-	-	0.09
76	C_15_H_24_	Copaene	-	-	-	-	-	-	-	-	-	0.14
77	C_15_H_24_	β-bisabolene	3.96	-	2.26	-	0.22	-	-	-	-	-
78	C_10_H_14_O	perillaldehyde	-	-	-	-	-	-	-	-	-	2.83
79	C_10_H_14_O_2_	Perillaketone	-	-	-	-	-	-	-	-	-	35.56
80	C_10_H_16_O	Myrtenol	-	-	0.02	0.83	-	-	-	-	-	-
81	C_10_H_14_O	α-α-4-trimethyl-Benzenemethanol	-	-	-	0.11	-	-	-	-	0.04	-
82	C_10_H_16_O	trans-Carveol	-	-	0.08	-	-	-	-	2.1	0.04	-
83	C_10_H_16_O	*cis*-Carveol	-	-	-	-	-	-	-	0.68	-	-
84	C_13_H_2_O	*trans*-α-bergamotene	-	1.9	-	-	14.16	-	-	-	-	-
85	C_15_H_24_	β-sesquiphellandrene	0.06	0.4	0.05	-	0.15	-	-	-	-	-
86	C_10_H_18_O	Nerol	0.06	-	-	-	0.17	-	-	-	-	0.07
87	C_10_H_12_O_2_	Dehydroelsholtzia ketone	-	-	-	-	-	-	-	-	-	0.59
88	C_10_H_12_O_2_	Isoegomaketone	-	-	-	-	-	-	-	-	-	20.4
89	C_12_H_16_O_2_	Thymol acetate	2.22	11.95	-	-	-	-	-	-	-	-
90	C_10_H_18_O	Geraniol	0.06	-	-	5.91	0.52	-	-	-	-	-
91	C_11_H_14_O_2_	Methyleugenol	-	-	-	-	1.53	-	-	-	-	0.27
92	C_15_H_26_O	Epicubenol	-	-	-	-	1.34	-	-	0.33	-	-
93	C_15_H_26_O	*trans*-Nerolidol	-	-	-	-	0.34	-	0.24	-	-	-
94	C_15_H_26_O	Viridiflorol	-	-	-	-	0.13	-	0.08	0.05	7.85	-
95	C_14_H_22_O	Norpatchoulenol	-	-	-	-	-	-	1.14	-	-	-
96	C_15_H_24_O	Caryophyllene Oxide	0.33	0.06	0.64	0.15	-	0.07	1.86	-	0.48	0.87
97	C_15_H_26_O	γ-Eudesmol	0.4	-	-	-	-	-	-	-	-	-
98	C_15_H_24_O	Espatulenol	0.03	-	0.53	-	-	0.03	0.03	0.01	0.04	0.22
99	C_15_H_26_O	Farnesol	-	-	-	-	-	-	0.9	-	-	-
100	C_12_H_16_O_4_	Pogostone	-	-	-	-	-	-	5.45	-	-	-
101	C_15_H_24_O	Humulene epoxide II	-	0.27	-	0.04	-	-	0.36	-	1.32	
102	C_10_H_12_O_2_	Eugenol	0.11	0.26	0.06	-	10.27	-	-	0.17	-	0.33
103	C_15_H_26_O	*τ*-Cadinol	-	-	-	-	7.98	0.02	-	0.07	-	0.03
104	C_12_H_16_O_3_	Elemicin	-	-	-	-	-	-	-	-	-	0.84
105	C_11_H_12_O_3_	Myristicin	-	-	-	-	-	-	-	-	-	0.11
106	C_12_H_16_O_3_	Isoelemicin	-	-	-	-	-	-	-	-	-	3.29
107	C_12_H_16_O_3_	Asarone	-	-	-	-	-	-	-	-	-	1.71
108	C_15_H_26_O	β-Eudesmol	-	-	-	-	0.45	-	-	-	-	-
109	C_15_H_26_O	Patchouli alcohol	-	-	0.01	-	-	0.4	50.52	-	-	-
110	C_15_H_26_O	Pogostol	-	-	-	-	-	-	5.43	-	-	-
111	C_10_H_14_O	Thymol	23.7	26.59	14.64	0.15	-	-	-	-	0.27	-
112	C_15_H_26_O	Neointermedeol	-	-	0.09	-	-	-	0.38	-	-	-
113	C_15_H_24_O	Longifolenaldehyde	-	-	-	-	-	-	0.27	-	-	-
114	C_12_H_16_O_2_	Carvacryl acetate	-	-	0.36	-	-	-	-	-	-	-
115	C_10_H_14_O	Pulespenone	-	-	1.75	0.15	-	-	-	-	-	-
116	C_11_H_16_O	*cis*-Jasmone	-	-	0.05	0.01	-	-	-	-	-	-
117	C_10_H_14_O	Carvacrol	2.48	7.66	20.82	0.07	-	0.16	-	0.14	0.05	-
118	C_15_H_26_O	α-Cadinol	0.03	-	0.03	-	0.26	-	-	0.32	-	-
119	C_12_H_14_O_3_	Eugenol acetate	-	-	-	-	0.36	-	-	-	-	-
120	C_14_H_22_O	2,4-Di-tert-butylphenol	0.07	0.05	-	0.03	0.04	0.03	-	-	0.04	-
121	C_20_H_40_O	Phytol	0.11	-	-	-	2.17	-	0.05	-	0.03	-
122	C_20_H_34_O	13-Epimanool	-	-	-	-	-	-	-	-	8.91	-
123	C_9_H_11_C_l3_NO_3_PS	Chlorpyrifos	-	-	-	-	0.47	-	-	-	-	-
Total Content (%)			90.51	94.25	89.87	86.68	92.06	97.02	81.58	96.85	97.24	94.22

**Table 2 molecules-25-04862-t002:** FRAP and DPPH values of 10 EOs (*n* = 6).

EOs	FRAP	DPPH (IC_50_, μg/mL)
TmEO	271.84 ± 4.93	1.42
McEO	480.66 ± 29.90	1.92
OvEO	633.71 ± 13.14	1.47
RoEO	22.14 ± 0.63	20.36
ObEO	1536.67 ± 24.22	<1
MhEO	22.32 ± 1.33	23.95
PcEO	30.35 ± 10.65	49.74
MsEO	24.54 ± 0.69	13.34
SoEO	17.22 ± 2.58	11.86
PfEO	67.14 ± 1.84	3.77
Trolox	2.02 ± 0.02	-
VC	-	6.04
PG	-	5.47

**Table 3 molecules-25-04862-t003:** Inhibition zone diameter (IZD, mm), MIC and MBC (μL/mL) for 10 EOs and positive control against four microorganisms (*n* = 6).

EOs	Activities	*S. aureus*	*E. coli*	*B. subtilis*	*S. enteriditis*
TmEO	IZD	30.77 ± 0.06	12.35 ± 0.73	11.59 ± 0.49	11.11 ± 0.19
	MIC	2	1	2	1
	MBC	4	2	2	1
McEO	IZD	32.32 ± 0.07	13.26 ± 0.45	14.72 ± 0.19	14.81 ± 1.55
	MIC	1	2	1	1
	MBC	2	4	1	2
OvEO	IZD	31.82 ± 0.03	10.78 ± 0.49	10.52 ± 0.29	10.71 ± 0.17
	MIC	2	4	2	1
	MBC	4	4	2	2
RoEO	IZD	6.00	6.65 ± 0.14	6.46 ± 0.12	6.67 ± 0.21
	MIC	4	8	8	4
	MBC	16	16	16	4
ObEO	IZD	12.98 ± 0.01	8.25 ± 0.17	6.82 ± 0.31	7.19 ± 0.50
	MIC	2	4	8	8
	MBC	16	16	16	16
MhEO	IZD	7.96 ± 0.34	8.20 ± 0.36	7.75 ± 0.20	7.94 ± 0.04
	MIC	16	NT	8	4
	MBC	16	NT	8	4
PcEO	IZD	12.02 ± 0.02	6.00	6.00	6.00
	MIC	NT	NT	NT	NT
	MBC	NT	NT	NT	NT
MsEO	IZD	9.78 ± 0.16	7.10 ± 0.08	6.93 ± 0.07	7.22 ± 0.04
	MIC	16	8	NT	NT
	MBC	32	8	NT	NT
SoEO	IZD	7.68 ± 0.39	6.00	6.00	6.00
	MIC	NT	NT	16	4
	MBC	NT	NT	32	8
PfEO	IZD	7.81 ± 0.46	6.00	6.00	6.44 ± 0.04
	MIC	32	NT	8	4
	MBC	NT	NT	16	8
CTC	IZD	16.12 ± 0.24	6.00	9.68 ± 0.12	6.00
	MIC	8	NT	NT	NT
	MBC	8	NT	NT	NT

NT: not tested; CTC: chlortetracycline hydrochloride. Filter paper diameter (6 mm) is included in the test results.

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
