# Peer review of "Chemical Composition and Bioactivity of Essential Oil of Ten Labiatae Species"

_molecules, 2020, doi:10.3390/molecules25204862_

Round 1
Reviewer 1 Report
The authors described the chemical composition, the antibacterial and antioxidant activity of ten essential oils of Labiatae species. The most interesting part of this study is the evaluation of the stability of the essential oils at different temperature and pH values and at the UV irradiation. In the results section, this part needs to be extended.

Author Response
Response to Reviewer 1 Comments
Point 1: In general in the manuscript write E.coli not E.Coli; S.aureus not S.a.; S.enteriditis not S.e.; B.subtilis not B.s.
Response 1: We have changed the abbreviations of the four microorganism.
Introduction section.
Point 2: Line 44: The reference 4 it is not adequate. The reference is about the antibacterial activity of bacteriocins, natural substances produced by the secondary metabolism of some bacteria. Bacteriocins and essential oils are very different from each other.
Response 2: A more appropriate article had been referenced.
[4] Guo SW, Ma JX, Xing YY, et al. Artemisia annua L. aqueous extract as an alternative to antibiotics improving growth performance and antioxidant function in broilers[J]. Italian Journal of Animal Science, 2020, 19(1):399-409. DOI: 10.1080/1828051X.2020.1745696.
Point 3: Lines 45-47: Please, write: Essential oils (EOs), important secondary metabolites of plants contain several chemical classes of compounds, which are volatile at normal temperature, immiscible with water and obtained by steam distillation from plants [5,6].
Response 3: Thank you, it has been modified. Line 46-48.
Point 4: Line 47: Please, write: EOs presented more biological properties, such…..
Response 4: Thank you, it has been modified. Line 48.
Results and discussion section:
Point 5: Line 83: Please, write identified, not indentified.
Response 5: Thank you, it has been modified. Line 83.
Point 6: Replace table 1 with table S1. Table S1 contains the chemical composition results.
Response 6: Thank you, we have replaced that table.
Chemical Composition section.
Point 7: Please, rewrite the considerations by comparing the various EOs with each other and with other authors’ studies.
Response 7: we have rewrite this paragraph. Line 87-112.
Point 8: Line 113: Please, delete it.
Response 8: We consider remaining it would be better.
Antibacterial activity section.
Point 9: Please, write Results of Inhibition Zone Diameter (IZD) and Results of Minimal Inhibitory Concentrations (MIC) and Minimal Bactericidal Concentration (MBC) Comparisons with the results of other authors are missing in both the Results of IZD and the Results of MIC and MBC sections.
Response 9: we have rewrite this paragraph. Line 163-198.
Point 10:Lines 144-152: The English language and grammar should be checked.
Response 10: we have rewrite this paragraph. Line 198-204.
Point 11: Effects of different pH on antibacterial stability of EOs section.
Please, rewrite by expanding the results with a comparison with the results of other authors.
The authors write that MhEO showed the strongest inhibitory effect on S.aureus, but also TmEO, McEo and OvEO. TmEO, 162 McEO and OvEO had the strongest inhibitory effect on E.coli at pH=6, but also a pH 7 and 8. The authors do not describe the inhibitory activity against B. subtilis.
Response 11: we have rewrite this paragraph. Line 213-222.
Point 12: Effects of different temperature on antibacterial stability of EOs section.
Please, rewrite by expanding the results with a comparison with the results of other authors.
Line 166: What is a normal condition? The room temperature or more probably 37°C?
Line 168: The authors write that there was no IZD of RoEO under normal conditions. In figure 4 and Table S3 RoEO showed a small antibacterial activity which is increased under high temperature.
Response 12: we have rewrite this paragraph. Line 224-231.
Point 13: In general, improve the legend of all figures.
Response 13: the questionable figures have been improved.
Materials and Methods section:
Point 14: Lack the Antibacterial stability of essential oils section. Please, insert it in the text.
Response 14: we have insert a new paragraph. Line 340-345.
Point 15: Line 225: What is the temperature of incubation? Please, add this information in the text.
Response 15: The temperature of incubation is room temperature, and it has been insert into the text. Line 278-288.
Point 16: Lines 236-237: FRAP reagent (180μL)….What is the concentration? In which solvent the reagent was dissolved? Please, add these information in the text.
Response 16: FRAP reagent was a kind of rapid test kits, buy from Shanghai Beyotime Biotechnolgy Co., LTD. And the volum of 180 μL is according to the operating instructions.
Point 17: Line 240: After 3-5 min. This is the incubation time? Usually, the samples added with FRAP reagent are incubated for 30 min at 37°C.
Response 17: It is the incubation time. And it is also according to the operating instructions.
Point 18: Line 249:Please, delete the double comma.
Response 18: It has been deleted.
Point 19: Line 249: cultured (37 °C) for 6 h after activated. Usually, the strains were incubated at 37 °C for 18-24h. Why the authors incubated the strains only for 6h? After 6h of incubation, the logarithmic phase of bacteria growth is not at its maximum.
Response 19: the right time is 24 h,and we have corrected it.
Point 20: Line 253: What kind of culture medium plate? Tryptic Soy Agar plates or Muller Hinton Agar plates? Please, insert it in the text.
Response 20: Muller Hinton Agar plates. Line 315.
Point 21: Line 254: How much is a bacteria solution? 100 μL? Please, insert it in the text.
Response 21: it is 100 μL. Line 316.
Point 22: Line 254: 2 pieces of paper contained EOs….2 pieces of paper because two different concentrations of EOs were used or because the test was performed in duplicate? Please, specify it in the text.
Response 22: The test was performed in duplicate.
Point 23: Lines 259-273: The English language and grammar should be checked. Please, specify that the tube11was the positive control and the tube 12 was the negative control. The times of strains and the plates incubation are low. Usually, the strains and the plates were incubated at 37 °C for 18-24h.
Response 23: we have rewrite this paragraph. Line 322-338.
Conclusion section:
Point 24: Please, extend this section.
Response 24: we have rewrite this paragraph. Line 352-359.
References section:
Point 25: The references section is not correctly written, please check it and write these following the Citation Style Guide present in the journal instructions
Response 25: All the references have been checked.
Reviewer 2 Report
The article is based on the study of the chromatographic characterization, antioxidant and antibacterial activity of essential oils from ten plants of the Labiatae family. The work could be of interest to Molecules readers, however, the quality of the writing and the lack of scientific discussion makes me doubtful about it.
In the abstract and in the introduction, the authors indicate that they will apply the essential oils in feed, however, nothing has been done about it.
In the abstract and in the introduction, the authors indicate that they will apply the essential oils in feed, however, nothing has been done about it.
The discussion is very confusing and poorly written. There is no link between the compounds found and the antioxidant and antibacterial properties of essential oils.
Significant English editing is needed.
In general, the article needs a thorough review of English. It is often not possible to understand what the authors would like to say.
Poorly written results and discussion with fragmented sentences. Poor discussion about the compounds found. Items and sub-items poorly outlined.
The methodologies on the effects of different pH, different temperature and effects of UV irradiation time on antibacterial stability of EOs have been overlooked. Only the results were shown without a discussion of the objectives of these methodologies.
Besides, the references are outside the journal's rules.
My comments and judgment are in the pdf file (highlighted in yellow).

Author Response
Thank you very much for the detailed comments, it help us a lot on the revising. now we have submited a in-depth revision.

Round 2
Reviewer 1 Report
The authors responded adequately to the reviewer comments.
Reviewer 2 Report
Dear
All recommendations were met and I believe that the final version has been improved. The critical paragraphs were rewritten and the information that was missing in the methodologies section was added to the text and all the references have been checked. In addition, the figures have been improved considerably. After the final edition, the manuscript can be published.